# Development of Assistance Level Adjustment Function for Variable Load on a Forearm-Supported Robotic Walker

**DOI:** 10.3390/s24196456

**Published:** 2024-10-06

**Authors:** Yuto Mori, Soichiro Yokoyama, Tomohisa Yamashita, Hidenori Kawamura, Masato Mori

**Affiliations:** 1Graduate School of Information Science and Technology, Hokkaido University, Kita 14, Nishi 9, Kita-ku, Sapporo 060-0814, Japan; 2Faculty of Information Science and Technology, Hokkaido University, Kita 14, Nishi 9, Kita-ku, Sapporo 060-0814, Japan; yokoyama@ist.hokudai.ac.jp (S.Y.); tomohisa@ist.hokudai.ac.jp (T.Y.); kawamura@ist.hokudai.ac.jp (H.K.); 3SUNCREER Co., Ltd., Sapporo 060-0012, Japan; mori@suncreer.co.jp

**Keywords:** robotic walker, walking assistance, smart walker, incremental PID, mobility aids, assistive robotics

## Abstract

With the progression of an aging society, the importance of walking assistance technology has been increasing. The research and development of robotic walkers for individuals requiring walking support are advancing. However, there was a problem that the conventional constant support amount did not satisfy the propulsion force required for the walking speed that users wanted. In this study, in order to solve this problem, we propose an algorithm for determining the support amount to maintain the walking speed when the average walking speed of each user is set as the target speed. A robotic walker was developed by attaching BLDC motors to an actual walker, along with a control algorithm for assistance based on sampling-type PID control. The effectiveness of the assistance determination algorithm and the usefulness of the parameters were demonstrated through experiments using weights loaded on the forearm support and target speeds. Subsequently, subject experiments were conducted to verify the ability to maintain target speeds, and a questionnaire survey confirmed that the assistance did not interfere with actual walking. The proposed algorithm for determining the assistance levels demonstrated the ability to maintain target speeds and allowed for adjustments in the necessary level of assistance.

## 1. Introduction

The importance of walking assistance technology is increasing globally as societies age. Walking is the most fundamental and crucial means of mobility for humans, closely linked to the quality of daily life [1]. However, it is not uncommon for walking ability to decline due to aging, illness, or accidents [2,3,4]. According to the “Summary Report of Comprehensive Survey of Living Conditions” [5] conducted by the Japanese Ministry of Health, Labour and Welfare in 2016, the leading cause of people needing nursing care is locomotor disorders (locomotive syndrome), which accounted for 24.6% of the total. Against this background, various walking aids such as canes, walkers, and wheelchairs have been developed for those needing walking assistance due to motor disabilities.

Recent technological advancements have led to the electrification and robotization of walking aids [6,7,8,9]. Walkers, compared to other walking aids, provide moderate activity for the lower limb muscles and adequate safety. In particular, the user is able to maintain their posture with greater ease as they are able to transfer a portion of their weight to the walker while standing, thereby sharing the load on their legs [10]. Robotic walkers (also known as Smart Walkers or Intelligent Walkers) have been developed to address some shortcomings of conventional walkers and provide more effective walking assistance. In the field of robotic walker research, motor-assisted propulsion is commonly implemented to reduce the walking load on users.

However, various approaches have been tested for how to drive the motor in response to inputs, including using handle-type input devices [11] or driving based on lower limb conditions [12]. Currently, there is no clear definition of the ideal form of assistance that robotic walkers should provide. Moreover, the ideal assistance varies for each user depending on their physical condition and desired walking speed. The quantitative evaluation of users’ sense of being assisted and ease of walking is complicated. One method of experimentation is to measure the electromyography of the lower limbs when walking, but there are difficulties regarding measurement and the complexity of interpreting the results.

In this study, we integrate sensor technology into a walker, a type of walking aid, and develop an algorithm to determine the amount of support needed to keep up with the user’s target walking speed. First, we present a simple physical model that illustrates how walking assistance is provided using a walker, detailing the forces exerted between the user, the walker, and the surrounding environment. Using this physical model, we clarify the variation in the required amount of assistance when walking with the support of a robotic walker. We then verify the achievement of the necessary amount of support from the robotic walker to maintain the desired walking speed using feedback control that adapts to these variations.

The purpose of this research is to develop an algorithm for determining the amount of assistance for robotic walkers based on physical quantities that do not depend on subject-specific evaluations. We suggest that an assistance determination algorithm based on mechanical analysis will provide more effective and consistent walking assistance compared to the conventional methods. This algorithm is determined by analyzing the free body diagram (a mechanical diagram illustrating all the forces acting on an object) of the forces acting on the robotic walker.

In this study, we develop a robotic walker with a wooden frame of the same shape as a commercially available forearm support four-wheeled walker and implemented the proposed algorithm. We confirm that a certain walking speed is maintained simulating different user body weights and forces imposed by the user in the forward direction by changing the load on the robotic walker. Subsequently, through subject experiments, we verify the ability to maintain speed with assistance and confirm that the user’s walking is not impeded by increases or decreases in the amount of assistance. Through this research, we aim to contribute to the optimization of robotic walker support systems and the development of new mobility aids to maintain and improve walking ability for more people who can walk using a walker.

This sensor-based technology promotes healthy aging and makes a significant contribution to supporting independence in rehabilitation and daily life. In addition, using the numerical data on walking collected by the sensors, it is possible to continuously monitor health conditions and detect abnormalities early using AI technology, which contribute to improving health.

The remainder of the paper is structured as follows: Section 2 presents the related research on the functions of conventional robotic walkers and methods for determining the amount of assistance. Section 3 describes the implementation method of the robotic walker used in the experiments, details of the proposed assistance amount algorithm, and the experimental procedures to prove that the assistance is achieved. Section 4 reports the results of load-specific experiments using weights and practical use experiments with subjects. Section 5 discusses each experiment from Section 4. Section 6 summarizes the proposed algorithm, experimental results, and discussion.

## 2. Related Research

The research and development of robotic walkers are rapidly progressing, aiming to support mobility for the elderly and physically disabled. Graf [13] developed a robotic walker that achieved obstacle avoidance behavior considering user input, verifying its ability to safely guide users to target locations in field tests. Jeong et al. [14] developed a robotic walker for daily life support, designed to assist with standing, sitting, walking, and toileting. They quantitatively verified the device’s effectiveness in various daily life usage scenarios. Our proposed method aims for a robotic walker that does not require a caregiver indoors, supporting actions from summoning the walker to one’s location to walking.

While it is important to ensure reliability in the control aspects used in robotic walkers, it is known that sufficient reliability can be ensured by using parts such as DC motors and single-board computers. DC motors are the primary actuators used to ensure propulsion in robotic walkers, and they continue to be employed in the latest robotic walkers [15,16]. For determining the level of assistance, walkers incorporating microcomputers such as Raspberry Pi have been developed [17,18]. This choice can be attributed to two main factors: firstly, the assistance determination algorithms adopted are sufficiently computable even on small devices; secondly, there is a concern regarding space optimization when integrating devices into the walker.

Some research has incorporated handle-type devices with grips as input devices for robotic walkers. The haptic interface mounted on the robotic walker developed by Morris et al. [11] is operated by gripping the handlebar. Pressure sensors are installed on the front and back of the handlebar, with forward pushing driving the motor forward, and a combination of push and pull differentials control the rotational drive. The robotic walker developed by Xiaoyang et al. [19] features a soft handle with built-in pressure sensors, used for detecting user operations and emergency events like falls. The walker’s speed for walking assistance is selected from five preset levels. These handle-based interfaces require unique operations not typically performed with conventional walkers. Non-contact information from devices like infrared cameras can also be used as an input. The robotic walker developed by Geunho et al. [12] uses a Laser Range Finder (LRF) to estimate the walking intentions from the user’s lower limb movements, determining the motor output to control the walker’s movement.

It is possible that the user may not reach the target walking speed with these control methods. Furthermore, there are still issues to be addressed in terms of achieving both stability and speed control in walking. We design our assistance algorithm focusing on walking speed, a common indicator of gait. People walk most efficiently in terms of energy and stability at their preferred walking speed [20]. Since the user’s walking speed follows the walker’s speed, measuring the robotic walker’s speed enables walking speed measurement without attaching devices to the person.

## 3. Materials and Methods

This section presents the functions that the proposed robotic walker should achieve and the methods to realize these functions.

### 3.1. Target Users and Usage Scenarios

The proposed robotic walker is designed for individuals who can walk independently with minimal assistance, corresponding to the Japanese Ministry of Health, Labour and Welfare’s care need levels of Support Required 1, 2, or Care Level 1.

The intended usage environment is indoor settings such as care facilities or homes. This is because the forearm support four-wheeled walker used in this study excels in providing stability on flat floors but is not suitable for outdoor use on uneven surfaces. If outdoor use were to be considered, a different type of walker designed for outdoor walking, such as a rollator, would be necessary. A key consideration for indoor use is the need for a caregiver to bring the walker to the user’s location. While keeping the walker constantly near the user might seem practical, this is not preferable. User interviews have revealed concerns about negative perceptions from others regarding the large space required for storage when not in use [21,22].

To address this, an autonomous navigation feature allows the walker to be delivered to the user’s location without caregiver assistance. Subsequently, the walker provides necessary propulsion assistance to reduce the physical burden of walking to the destination. This research focuses on the propulsion assistance function used to alleviate the strain of walking rather than the autonomous navigation aspect.

To realize the presented scenario, the robotic walker requires the following functions:Mobility while supporting a portion of the user’s body weight;Autonomous navigation to the destination;Reduction in the user’s walking load;Collision prevention during walker use.

### 3.2. Forces Generated When Walking with a Robotic Walker

The support required by users of robotic walkers is a complex element that changes dynamically. This support varies moment by moment based on the force applied to the walker by the user and the walking speed, and needs to be appropriately adjusted according to the user’s walking state. In this research, we define the ideal support amount as “the motor torque required to maintain the user’s preferred walking speed, allowing for sustained and efficient walking”.

The method of providing support varies widely depending on the design of the robotic walker. Walkers equipped with input devices like handles detect pressure or tilt transmitted to these devices to determine the amount of support provided by motors mounted on the tires. On the other hand, walkers without input devices often provide a fixed amount of support in the direction of movement. However, the actual support required during walking is a complex function that changes dynamically based on the relationship between motor speed and torque, as well as the force applied by the user to the walker.

To understand the main force relationships that occur when walking with a forearm support walker, Figure 1 shows a side view of the user and the robotic walker. This force distribution is influenced by various factors such as the material of the floor and frame, and the position where the user places their forearms. The focus of this research is on identifying variables that determine the ideal amount of support during walking rather than detailed physical modeling. To simplify this analysis and extract the essential force relationships, we set the following conditions for illustration:The rolling friction force is equal for all tires;The weight on all tires is equal;There is no forward or backward slippage between the user’s forearm and the support platform.

To understand the mechanical behavior of the robotic walker, we define each physical quantity using the coordinate system shown in Figure 1. The right direction is defined as forward, with physical quantities in this direction being positive, while the left direction is backward, assigned negative signs. The horizontal velocity *v* of the walker typically falls within the range of ±1.5 m/s considering walking speeds. Air resistance acting on the walker is negligible compared to other forces and is not considered in this analysis. The main forces occurring during walking include the ground reaction force, which can be decomposed into $F_{v}$ and $F_{h}$. $F_{v}$ represents the vertical force for body weight support, while $F_{h}$ represents the horizontal force for propulsion. Additionally, $F_{y}$ is the force generated by placing the forearm on the forearm support, which expands the support base and improves postural stability. An increase in $F_{y}$ leads to a decrease in $F_{v}$, reducing the burden on antigravity muscles. An important force affecting the walker’s movement is the rolling friction force $f_{r}$, which occurs at the contact surface between the tires and the floor. It is expressed by the following equation: (1)$$f_{r}=\mu_{r}\left(mg+F_{y}\right)$$
where $\mu_{r}$ is the rolling friction coefficient, *m* is the mass of the walker, and *g* is the gravitational acceleration. The BLDC motor, which is the source of propulsion, converts electrical energy into mechanical energy. The torque generated by the motor provides rotational force to the wheel, which is converted into propulsive force $f_{m}$ through friction with the ground. This relationship is expressed by the following equation: (2)$$f_{m}=kP/v$$
where *P* is the controlled power, *v* is the velocity, and *k* is a coefficient indicating the motor’s conversion efficiency. The walker used in this experiment is rear-wheel drive, with the motor’s propulsive force acting only on the rear wheels. The power *P* is adjusted by changing the duty cycle through PWM control.

These relationships provide insights into the mechanical behavior when using a robotic walker. Equation (Equation 1) shows that increasing $F_{y}$ for body weight support also increases the rolling friction force fr in the opposite direction of movement, requiring greater propulsive force. Equation (Equation 2) indicates that the motor’s torque changes based on the current walking speed *v*. These analysis results reveal that the support required during walking changes dynamically according to the state of the user and the walker. Therefore, this research aims to implement a support adjustment function that adapts to this varying support requirement and maintains the ideal walking speed.

### 3.3. Walker Frame

This research aims to enhance the functionality of a forearm support walker by incorporating devices, sensors, and actuators. We constructed a wooden frame modeled after a commercially available four-wheeled forearm support walker. The front wheels are equipped with swivel casters, while the rear wheels feature fixed-direction tires with built-in BLDC motors. A forearm support four-wheeled walker is a type of walking aid where the user places their forearms, bent at a 90° angle, on the support platform to bear their weight while using the device. In this paper, the term “walker” refers to this forearm support four-wheeled walker. A notable characteristic of this walker is its superior stability during walking compared to other walker designs. The wooden frame has been constructed by a furniture workshop with experience in the production of wooden wheelchairs and similar items. As a result, the durability and reliability required for this verification have been assured. Figure 2 and Figure 3 show the exterior of our proposed robotic walker, equipped with the necessary sensors and actuators to fulfill its intended functions. The microcontroller and drivers, which will be explained in Section 3.4, are mounted on the side of the frame.

### 3.4. Sensors and Actuators

The walker developed in this research is equipped with four main functions: body weight support, autonomous navigation, walking load reduction, and collision prevention. Table 1 shows the devices installed to realize these functions. The body weight support function adopts the same shape as commercially available forearm support walkers, already meeting the necessary requirements. The autonomous navigation function is achieved through a combination of hardware and software. Hardware-wise, a ZED2 stereo camera mounted on the front of the walker and an NVIDIA Jetson Nano are used, while the software implements the Navigation Stack, an ROS package. The walking load reduction function is achieved by controlling the motor torque through the assistance amount determination algorithm proposed in Section 3.5. For collision prevention, a method using depth images obtained from the ZED2 to determine avoidance actions has been adopted, with obstacle avoidance performance against walls already verified [23]. The core of motor control is an Arduino Uno. This microcontroller is connected to BLDC motor drivers and controls the rotation direction and torque of the left and right wheels. Torque control is achieved by varying the voltage applied to the BLDC motors. Specifically, the voltage is controlled by adjusting the duty cycle in the PWM output of the Arduino Uno. The PWM signal frequency is set to 490 Hz, with a 0% duty cycle outputting 0 V and a 100% duty cycle outputting 5 V. On the Arduino, the duty cycle is defined as an integer value from 0 to 255 [24]. By varying this value, fine adjustments to the motor assistance can be made. Figure 4 shows the data flow between these devices. This configuration enables the realization of safe and effective robotic walker functions.

### 3.5. Ideal Support and Adopted Support Volume Algorithm

In this research, ideal support refers to assistance that allows users to maintain their individual ideal walking speed regardless of the force they apply to the walker. According to Holt et al.’s research [20], people optimize their energy efficiency and stability when walking at their preferred speed. To achieve this ideal support, precise motor control of the robotic walker is crucial. Specifically, by finely adjusting the speed of the left and right tires, not only straight movement but also curved direction changes become possible. Furthermore, in navigation to the destination, predicting the shortest route typically chosen by humans and identifying tire speeds based on this can provide more natural and efficient walking support.

To realize this ideal support, we developed a motor control system utilizing incremental PID control. This control system enables precise and adaptive speed adjustment according to the user’s walking characteristics. The adoption of incremental PID control was driven by the need to reduce computational demands when performing calculations on the Arduino Uno, thereby minimizing response delays. Moreover, it offers the advantage of maintaining stable control even when the speed measurements obtained from the Hall sensor contain outliers. The control algorithm is expressed by the following equations: (3)$$MV_{n}=MV_{n-1}+∆MV_{n}$$
(4)$$∆MV_{n}=Kp\left(e_{n}-e_{n-1}\right)+Kie_{n}+Kd\left(\left(e_{n}-e_{n-1}\right)-\left(e_{n-1}-e_{n-2}\right)\right)$$

Here, the control variable is the PWM duty ratio that enables the increase or decrease in current supplied to the motor. The target value is the ideal walking speed set for each user, and the input is the actual speed calculated from the Hall sensor attached to the motor. $MV_{n}$ represents the current (time *n*) duty ratio, and $MV_{n-1}$ is the duty ratio from the previous step. $e_{n}$ is the deviation between the target speed and current speed, and $e_{n-1}$ is the deviation from the previous step. Kp, Ki, and Kd are the coefficients for the proportional, integral, and derivative terms, respectively. To accommodate walker movements such as direction changes, the left and right motors are controlled individually. The parameters were calibrated in a real-world setting rather than a simulated one. Consequently, the primary auto-tuning techniques were not employed due to constraints such as the maximum driving distance and floor surface area of the experimental environment. Each control coefficient was manually tuned and determined as Kp = 0.03, Ki = 0.08, and Kd = 0.06. This manual tuning was based on the following criteria:No oscillation occurs when the speed stabilizes at the ideal speed.Even when the load on the robotic walker fluctuates, the maximum speed due to overshoot does not exceed 1.5 times the ideal speed.

Particular emphasis was placed on suppressing oscillations during speed stabilization. Despite the challenge of eliminating overshoot entirely, we were able to achieve a level within an acceptable range. The speed measurement using the Hall sensor adopted this time requires a minimum rotation angle of 51.43°. When starting up from a stop, fixed duty ratio torque control is used, and, when speed measurement becomes possible, the method of transitioning to PID control is adopted. Minimization of settling time was not considered in this tuning because the acceleration at the start of walking varies greatly depending on the user and situation. The upper limit for the duty ratio of the control amount was set at 62.75%. It was determined that, if a propulsive force exceeding this upper limit were applied, the user would receive excessive propulsive force assistance from the robot walker, which would result in the collapse of the normal stable walking posture.

## 4. Results

### 4.1. Performance Evaluation of the Proposed Support Quantity Algorithm under Load Variation

An experiment was conducted to ascertain whether the control algorithm for the proposed support amount could maintain a constant speed when several weights were placed on the forearm support.

#### 4.1.1. Purpose

The principal objective of this experiment was to ascertain the capacity of the developed robotic walker’s control system to maintain a constant speed in the presence of varying load conditions on the forearm support. The force applied by the user to the forearm support of a walker varies over time, depending on factors such as posture. In this experiment, we will assume a load that does not change over time. By conducting experiments under several load conditions, we will attempt to approximate the same load conditions as would be experienced in the real environment. This was conducted to evaluate the robustness and adaptability of the control system. The experiment was designed to test whether the control algorithm could effectively respond to changes in the force applied to the robotic walker and maintain the desired speed. Additionally, by examining the system’s responsiveness and stability under different load conditions, we aimed to comprehensively assess the control system’s performance in real-world scenarios. The objective was to identify an optimal control method for assisting the walking of elderly individuals.

#### 4.1.2. Experimental Setup and Procedure

The experimental setup was designed to emulate actual usage conditions as closely as possible. Two speed settings were adopted based on the medium-scale walking experiment data by Hao et al. [25]. The average walking speed of elderly individuals who regularly use walkers was 0.46 m/s, while the average speed of those who do not regularly use walkers was 0.79 m/s. To reproduce load conditions, the forearm support of the robot walker was loaded with weights of varying masses, and trials were conducted. For a user weighing 65 kg, the maximum load was set at approximately 20 kg, which represents approximately 30% of the user’s body weight. The loaded weights were set at three levels: 8 kg, 14 kg, and 20 kg, with experiments conducted for each weight condition. In consideration of actual use, it is acknowledged that there will be differences in weight between users. However, it is expected that a propulsive force will be gained from walking on the floor and the soles of the feet. Consequently, the load weight of 20 kg was selected as a weight that would ensure that the proposed algorithm would be able to maintain the walking speed even if the user supported their weight on the walker excessively. The load weight of 20 kg is predicated on the assumption that the user will be solely reliant on the forearms of the walker for support and will not be utilizing their legs for propulsion. Assuming a rolling friction coefficient of 0.054, which is an approximate value, a load weight of 14 kg reproduces the condition where the assumed user applies their weight to the forearm support of the walker and also applies a force of approximately 3.17 N in the forward direction, according to Equation (Equation 1). Similarly, the load weight of 8 kg reproduces the situation where the assumed user adds their body weight to the forearm support of the walker and also applies a force of approximately 6.34 N in the forward direction. This method enabled the simulation of user propulsion force variation through the increase in the rolling friction force acting on the wheels and the generation of a force opposite to the direction of propulsion. The experiments were conducted on a 12-m straight course set up on a flat indoor floor. A one-meter buffer zone was established at the end of the course to eliminate the effects of deceleration, with the objective of focusing on performance evaluation during acceleration and steady-state conditions. For the purpose of data collection, the values used for the input, output, and control variables were simultaneously recorded in order to confirm that the PID control was functioning correctly. Specifically, we acquired data on instantaneous speed from Hall sensors, on PWM duty ratios adjusted by PID control, and on self-localization data from stereo cameras. This multifaceted data collection enabled detailed observation of system behavior. The analysis results were presented in graph format to facilitate intuitive understanding of the system’s behavior under each load condition.

#### 4.1.3. Experimental Results

This section presents the findings of the performance evaluation experiment conducted on the proposed support amount determination algorithm. The experiment served to verify the algorithm’s capacity to maintain speed under varying load conditions. The robot walker is equipped with a Hall sensor on each of the left and right BLDC motors. However, due to the absence of a notable discrepancy in the recorded values of the two sensors, the data obtained from the Hall sensor on the right tire were deemed sufficient for analysis. Five trials were conducted for each parameter setting; however, no significant differences were observed between the trials, and, thus, one representative result is presented for each parameter. Figure 5, Figure 6, Figure 7, Figure 8, Figure 9 and Figure 10 illustrate the measured values of the sensor and actuator under varying parameter settings. The vertical axes of these graphs depict the speed calculated by the Hall sensor, the speed calculated from the stereo camera’s self-position estimation, and the PWM duty ratio adjusted by the control algorithm. The horizontal axes depict the elapsed time, with 0 representing the instant at which torque was generated in the motor. As evidenced in the results presented in Figure 5, Figure 6 and Figure 7, wherein the target speed was set to 0.46 m/s, it was ascertained that the speed was maintained at the target value without substantial vibration following the completion of acceleration. Nevertheless, instances of overshooting, defined as a temporary exceedance of the target speed, were observed during the acceleration phase. The greatest degree of overshooting was observed in conjunction with the lowest load and speed (Figure 5). However, the maximum speed did not exceed 1.5 times the target speed. Figure 8, Figure 9 and Figure 10 illustrate the effectiveness of the algorithm in maintaining the target speed of 0.79 m/s. However, in the case of a target speed of 0.79 m/s, a phenomenon was observed whereby the duty ratio reached its maximum value of 62.75% during acceleration, irrespective of the load. It was demonstrated that the speed calculated from the Hall sensor exhibited a delay of approximately two seconds in comparison to the speed calculated from the self-position estimation. Moreover, the maximum value of the duty ratio exhibited a discernible change under each condition, indicating that the requisite torque to attain a constant speed increased in proportion to the load. These findings demonstrate that the proposed assistance amount determination algorithm is capable of maintaining a constant speed despite load fluctuations. Additionally, insights were gained regarding the response characteristics of the control system under varying load conditions and the measurement characteristics of the sensor.

### 4.2. Walking Experiment Using the Proposed Support Amount Algorithm

The objective of the proposed support amount control algorithm is to ensure that the robot does not provide support that differs from the user’s walking intention. To this end, walking experiments will be conducted to confirm that the algorithm does not add support that differs from the user’s walking intention when the user walks using the robot walker.

#### 4.2.1. Purpose

The principal objective of this experiment is to assess the practicality and safety of the proposed control algorithm. In particular, the objective is to ascertain whether the algorithm can successfully diminish the physical burden on the user while maintaining their natural gait when utilizing a walker. The impact of enhancing the user’s mobility, a primary objective of robotic walkers, is challenging to quantify due to its subjective nature. Moreover, given the variability in individual walking patterns, the optimal level of assistance is contingent upon the specific individual. In previous studies, electromyography was employed on occasion. However, as Cram [26] notes, there are several challenges associated with this approach, including the difficulty of accurately positioning electrodes and the complexity of interpreting the resulting data, as highlighted by Disselhorst-Klug et al. [27]. In view of these considerations, this study concentrates on more generalizable evaluation metrics, such as irregular changes in speed and the presence of an excessive load in relation to the user’s intention. This approach enables an objective and comprehensive evaluation that is less susceptible to individual differences and subjective factors. In particular, we employ an analytical approach that integrates data from measurements with responses from questionnaires completed by the subjects. This method enables a comprehensive evaluation of the system’s stability and safety during walking.

#### 4.2.2. Experimental Setup and Procedure

To substantiate the security of the algorithmic output with respect to assistance and ambulatory stability, this experiment involved five subjects in their 20 s with a minimal risk of falling and no history of lower limb concerns. Prior to the commencement of the experiment, the subjects were provided with comprehensive information regarding the research purpose, potential risks, and the requirements of informed consent, in accordance with the guidelines set forth in the Helsinki Declaration. The subjects were then asked to indicate their agreement to participate in the experiment. Furthermore, the questionnaire items presented in Table 2 were elucidated prior to the commencement of the experiment. Given that the age group of the subjects is younger than that of the intended users, an additional 10 kg vest-type weight and 1 kg weights on each ankle were attached to increase the walking load. Figure 11 shows the external appearance with attached weights. This was conducted in order to more faithfully reproduce the walking conditions of elderly or mobility-impaired individuals. The experimental procedure was as follows:Following an explanation of the basic operational principles of the walker, the subjects were permitted a minimum of five minutes of unstructured walking time to become acquainted with the device.The steady walking speed of each subject was measured and established as their individual target speed. The measurement was based on the average speed of the central portion (4 s) between the acceleration start and stop as the target speed for the 12-m straight course. The same measurement method was employed to determine the target speeds for the left and right motors on the circular course.The walker, equipped with the proposed control algorithm, was used to traverse two distinct types of courses.(a)12 m straight course (6 trials);(b)Circular course with a 2 m radius, 2 laps (3 trials).

During each trial, the following data were recorded: walking speed calculated from self-position estimation by stereo camera, speed obtained from BLDC motor Hall sensors, and PWM duty ratio. From these measurement results, we verify that there are no fluctuations in speed that the user did not intend, and that there are no excessive numerical fluctuations (defined as vibrations of 5% or more per second) in the duty ratio determined. Following the completion of all courses, a questionnaire was administered, based on the items in Table 2, which also invited free-form opinions for each trial. This experimental design is intended to evaluate the practicality and safety of the proposed algorithm, and to gain important insights for future clinical applications.

#### 4.2.3. Experimental Results

The objective of this section is to elucidate, through a subject experiment, that the control algorithm for the support amount, as designed in Section 3.5, is capable of maintaining a steady state speed without placing any additional load on the actual user’s walking.

The results of the measurement of each subject’s walking speed in a steady state are shown in Table 3. The steady state was defined as the speed during the middle part (4 s) of the period from the start of acceleration to the stop, and the section where the variation in walking speed was minimal was adopted. From the measurement results, the average walking speed of the subjects in this experiment was within the speed range (0.46 m/s–0.79 m/s) assumed in the experiment using the weights in Section 4.1.2.

The results of the 12-m straight-line walking experiment demonstrated three primary trends in the relationship between speed variation and duty ratio. Figure 12, Figure 13 and Figure 14 illustrate representative patterns of these. The vertical axis of each graph depicts the speed as measured by the Hall sensor, the speed as estimated by the stereo camera, and the PWM duty ratio, while the horizontal axis represents the elapsed time from the point at which the motor torque was generated. Figure 12 (Subject A) illustrates a pattern wherein the speed stabilized at the target speed, concomitant with a stabilization of the duty ratio. Figure 13 (Subject C) depicts a scenario wherein the walking speed surpassed the pre-established steady-state speed, accompanied by a discernible tendency for the motor torque to decline gradually. Figure 14 (Subject E) illustrates a gradual increase in torque at the commencement of walking, followed by a phenomenon of the motor reaching maximum torque and subsequently decelerating to achieve the ideal walking speed. In all the trials conducted with all the subjects, there were no significant deviations from the intended route, and no sudden acceleration or dangerous changes in speed were observed. In each subject’s trial, a consistent tendency was observed whereby the walking speed stabilized around the target speed.

The results for the two-lap course with a two-meter radius exhibited a comparable pattern to that observed in straight-ahead walking. The results for the lap route will be presented as representative examples. Figure 15 illustrates the speed change and duty ratio for a single trial conducted by Subject C. The vertical axis of the graph depicts the speed measured by the left and right Hall sensors and the PWM duty ratio, while the horizontal axis represents the elapsed time from the point at which the motor torque was generated. In the case of Subject C, the optimal speed for a 2 m radius was found to be 0.504 m/s for the left tire and 0.714 m/s for the right tire. The experimental results demonstrated that these specific speeds were generally maintained during walking.

A questionnaire was employed to ascertain whether any load could not be observed by the sensors installed in the robot walker. This was conducted after all the trials had been completed. The results of the subjects’ answers to the questions in Table 2 are presented in Table 4. For Q1, four subjects answered “Totally Agree” and one answered “Agree”. For Q2, three subjects answered “Totally Agree” and two answered “Agree”. For Q3, three subjects answered “Totally Agree”, one answered “Agree”, and one answered “Neutral”. For Q4, three subjects answered “Totally Agree” and two answered “Agree”. The following responses were obtained from the open-ended questions about overall support and each trial.

I felt the force in an unintended direction only once when walking on the circular course in one trial (Subject A);I did not feel much support when walking on the circular course (Subject B);I felt that it would be okay to increase the amount of support even a little when walking in a straight line (Subject B);I felt the motor fluctuate depending on the speed, but it was not a change where the walker suddenly pulled me (Subject C);It was difficult to tell whether it was getting easier or not (Subject C);I felt that the strength of the motor was appropriate (Subject D);I felt that the walker was moving forward (Subject E).

## 5. Discussion

The results of the experiments on the load-dependent support amount determination algorithm, as detailed in Section 4.1.3, indicate that the control system of the robot walker has the capacity to maintain an appropriate speed. This characteristic is adaptable regardless of the degree to which the walker bears the user’s body weight. This represents a significant advancement towards the creation of a walking support system that can adaptively respond to the diverse needs of users. Conversely, the overshoot observed during acceleration underscored the inherent trade-off between the control system’s responsiveness and stability. In this manual tuning of PID parameters, adjustments were completed to bring the rise time closer to the acceleration time during actual walking, and to reduce excessive speed due to overshoot as much as possible. If the load is fixed, it is possible to approximate the actual walking speed data by adjusting the PID control parameters appropriately. However, the force applied to the forearm support of the robot walker differs depending on the user, and it is difficult to maintain this approximation for multiple loads with the sensors used in this experiment. One factor is that the inertia moment changes depending on the load, and the amount of torque required for acceleration changes. One way to solve these problems is to use a weight sensor to measure the load on the robot walker and dynamically adjust the PID control parameters according to the weight. In addition, to reduce response delays, it is also necessary to consider improving the performance of the devices used as computers. When changing devices, it is necessary to analyze in detail which processes are causing the time bottlenecks in determining the amount of support.

The delay in the Hall sensor measurements (approximately two seconds) unambiguously demonstrated the challenge in speed control in the low-speed range. This delay presents a challenge when attempting to gain propulsion during acceleration. To address this issue, it would be beneficial to consider the introduction of sensor fusion technology, which combines data from the stereo camera and Hall sensor, or the adoption of more sensitive sensors. The observed increase in torque requirements due to the increase in load is an important result that demonstrates the adaptability of the control system. By ensuring the ability to maintain speed in these cases with different loads, it is possible to customize the system to suit the characteristics of each individual user and their usage environment, thereby achieving more advanced personalized support.

The efficacy and reliability of the system when utilized by users in authentic scenarios were substantiated by the experimental outcomes presented in Section 4.2.3. The primary outcome was that the algorithm exhibited consistent and reliable speed maintenance capabilities in real-world usage scenarios. Nevertheless, a number of issues were also identified.

The observed differences in steady-state walking speed among the subjects indicate the diversity of individual walking characteristics. This result indicates the necessity for a control system for a robotic walker that is capable of adapting flexibly to the walking patterns of individual users. It will be necessary in the future to develop an adaptive control system that learns the walking characteristics of the user and optimizes them over time. In particular, if we can identify the gait cycle using motion tracking technology, it will become possible to adjust the amount of assistance according to the cycle. Specifically, since elderly individuals tend to have a reduced ability to push off with their feet during the swing phase of walking [28], a control system that varies the amount of assistance during specific gait phases can be considered. The three types of speed-duty ratio patterns observed in the straight-line walking experiment demonstrated the adaptability and robustness of the proposed algorithm. In particular, the stabilization of speed in proximity to the target speed observed in all the subjects indicates a high degree of capability to maintain speed during the act of steady walking. However, further investigation is required to elucidate the reasons behind the deviation from the target speed observed in some trials, particularly the phenomenon observed in Figure 13.

The observation in Figure 13, wherein the user maintains a slightly elevated speed relative to the target speed and exhibits no deceleration despite the reduction in torque, indicates a potential interaction between the user’s gait and the algorithmic control. This phenomenon suggests that the user, who is aware of maintaining speed, may be increasing the propulsion force gradually unconsciously. To substantiate this hypothesis, a more comprehensive biomechanical analysis and prolonged walking experiments are necessary. One of the limitations of this study is that it was not feasible to continue walking until the torque stabilized completely due to the restricted length of the experiment (12 m). Potential solutions to this issue include conducting longer walking experiments and long-term observation of steady-state walking using a treadmill. Conversely, enhancing the responsiveness of speed adjustment by expanding the scope of torque fluctuations is a potential avenue for improvement. However, there is a risk that abrupt alterations in the level of support may have a detrimental impact on the user’s posture control or, alternatively, may result in a reduction in the capacity to maintain speed. The optimization of this trade-off represents a significant research opportunity for the future.

Another limitation of the algorithm is its inability to adapt to situations where the user’s target walking speed changes. For example, when walking while conversing with another person, the user needs to match the walking speed of their companion. As a result, the target speed differs from the user’s usual ideal walking speed. In such cases, the current algorithm cannot accommodate these variations.

From the perspective of safety, it is noteworthy that no sudden acceleration or dangerous changes in speed were observed throughout all the trials. In addition, the proposed support amount determination algorithm did not fluctuate the support amount suddenly, and it did not affect the user’s walking posture. The results demonstrate that the proposed algorithm is capable of providing effective support while ensuring the safety of walking. Nevertheless, it is imperative to conduct further verification and establish this safety through long-term clinical trials with actual elderly individuals and those with heavy walking loads. In this validation, to confirm that the proposed algorithm can track the subjects’ target speeds, we conducted tests on five individuals with a low risk of falling during walking. To approximate the walking conditions of the intended users, measurements were taken under increased walking load by attaching additional weights. Ensuring safety in real-world environments will require a larger and more diverse sample, including elderly individuals and those with mild motor impairments. In addition, in the parameter adjustment for this study, it was confirmed from Figure 5, Figure 6, Figure 7, Figure 8, Figure 9 and Figure 10 that there is no steady-state error for short-term walking of less than 12 m. However, since incremental PID control is employed, there is a possibility that an integral truncation effect may occur. Therefore, additional verification regarding the accumulation of long-term errors is necessary.

## 6. Conclusions

This research addresses the issue of determining the optimal amount of support in response to fluctuations in load weight when using a robotic walker. To this end, a robotic walker with a speed maintenance function has been developed using sampling-type PID control. The findings establish a foundation for walking support technology that adaptively responds to physical attributes and the user’s walking speed.

The main scientific achievement was the proposal of an algorithm for determining the amount of assistance to provide to a robotic walker in order to follow a target speed. Multiple experiments showed that the algorithm was effective as a propulsion force assist without interfering with walking. In the experiment, weights were loaded to simulate the load on the forearm support of the user, and the speed maintenance ability and system stability were verified under the average speed conditions of the assumed user. Consequently, precise control of the torque amount was achieved through the implementation of PID control, resulting in the maintenance of a stable target speed. It is noteworthy that the overshoot was suppressed to within 1.5 times the ideal speed, which is an important result that demonstrates both safety and comfort.

A walking experiment was conducted with five subjects in their 20 s, who were deemed to be at low risk of falling when walking. The proposed support amount determination algorithm was thus demonstrated to be a practical solution. In the experimental design, the discrepancy in the walking load between the assumed user and the experimenter was considered, and the efficacy of the algorithm was evaluated under more realistic usage conditions by attaching a 12 kg weight. Consequently, the system was capable of dynamically adjusting the amount of assistance by the subject’s individual steady-state walking speed, thereby demonstrating the algorithm’s high degree of adaptability. The findings of the questionnaire survey are in alignment with the assumption that the developed system is capable of supporting natural walking movements without impeding the gait of the user. On the other hand, the tendency to maintain a walking speed faster than the steady state observed in some trials presents a topic for further investigation. This phenomenon indicates the potential for an unconscious increase in propulsion force by the user, underscoring the necessity for a more comprehensive biomechanical analysis and long-term use evaluation. Potential solutions to this issue include enhancing the responsiveness of torque and integrating intentional speed reduction assistance. However, the trade-off relationship with speed maintenance ability must be carefully considered.

The findings of this study indicate that an algorithmic approach has been proposed for determining the optimal level of support to be provided, with the objective of adapting to the individual’s unique walking characteristics and ensuring the provision of safe and effective walking support.

In order to facilitate its practical implementation, it is imperative to subject it to further verification under a more diverse range of conditions, evaluate its long-term usability, and establish its efficacy through clinical trials. One improvement to the algorithm proposed in this study is the introduction of a mechanism to automatically estimate the target speed. In this verification, we confirmed whether the proposed support amount determination algorithm can follow the target speed without interfering with walking when the target speed is determined. When considering the actual walking environment, the walking speed that users want may differ from their everyday walking speed depending on the surrounding situation. To realize this function, it is thought that the introduction of advanced object detection technology using 360° LiDAR sensors and multiple cameras is possible. This would make it possible to grasp the surrounding situation in real time and approach the optimal walking speed. However, there are also issues with this improvement. The introduction of advanced sensor technology and object detection algorithms requires a large amount of computing power, and it is inevitable that sufficient power will be secured and the size of the device will increase. Furthermore, the accumulation of walking data using wearable devices and the cloud is also an important issue. By integrating these, more adaptive walking support that takes into account the user’s health condition and activity level is expected.

## Figures and Tables

**Figure 1 sensors-24-06456-f001:**
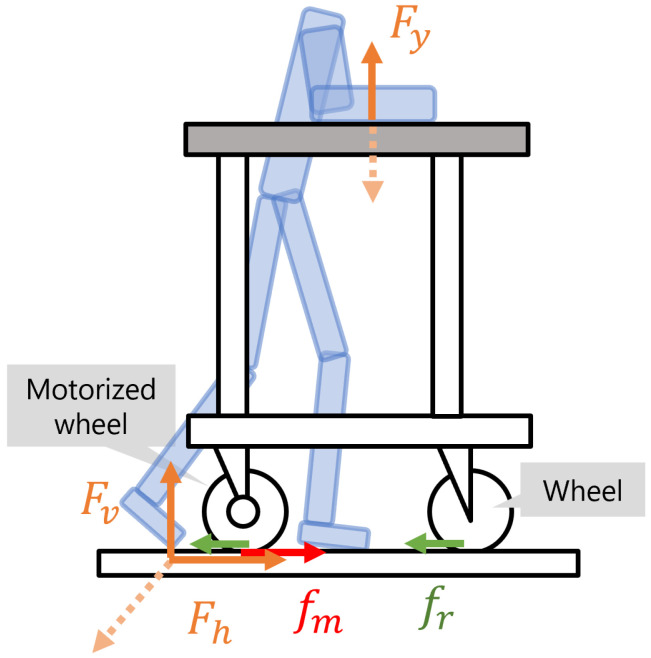
Main forces when walking with a forearm-supported walker.

**Figure 2 sensors-24-06456-f002:**
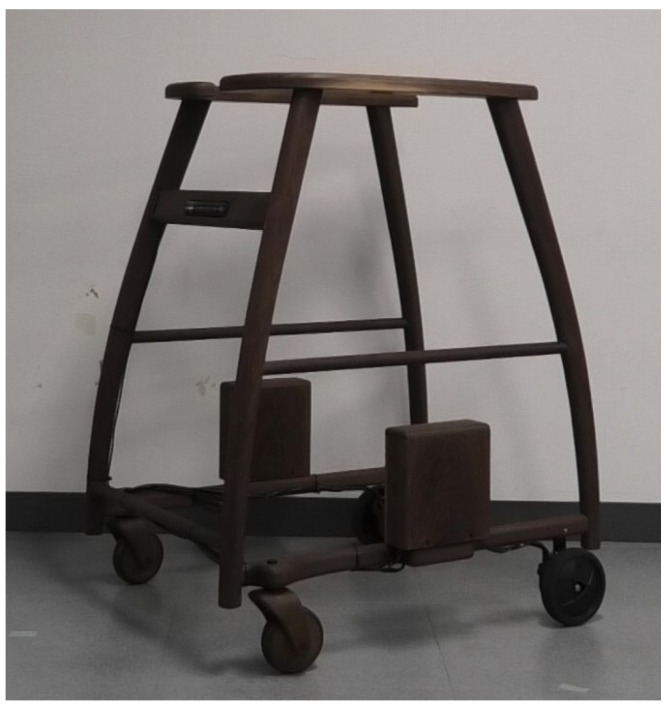
Appearance of the robotic walker when not in use.

**Figure 3 sensors-24-06456-f003:**
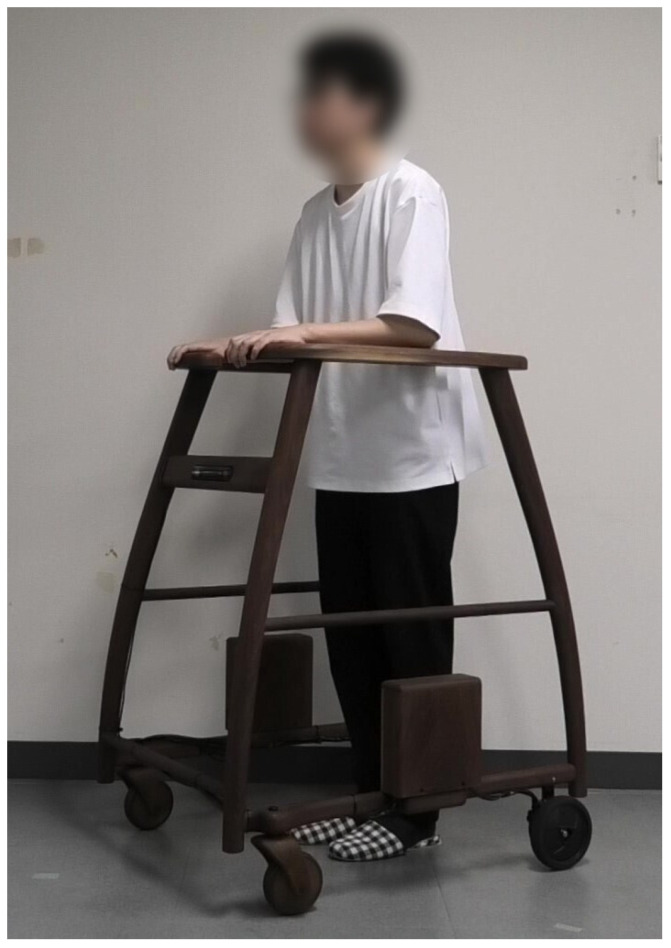
Appearance of the robotic walker when in use.

**Figure 4 sensors-24-06456-f004:**
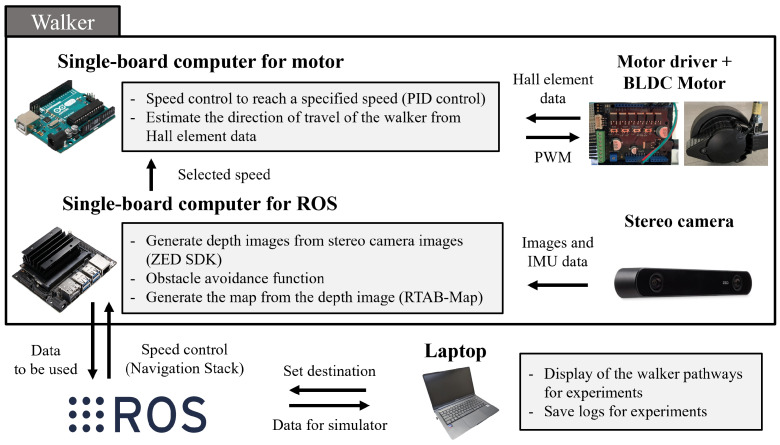
Architecture of equipment on a robotic walker.

**Figure 5 sensors-24-06456-f005:**
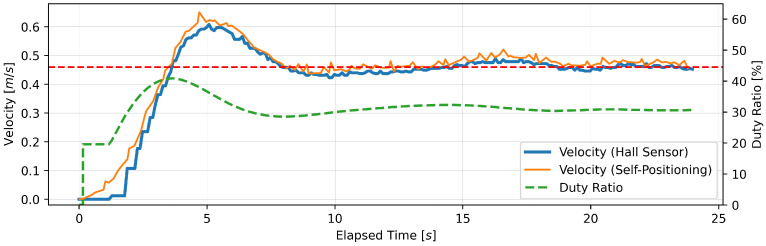
Output of speed and duty ratio at a load capacity of 8 kg and a target speed of 0.46 m/s.

**Figure 6 sensors-24-06456-f006:**
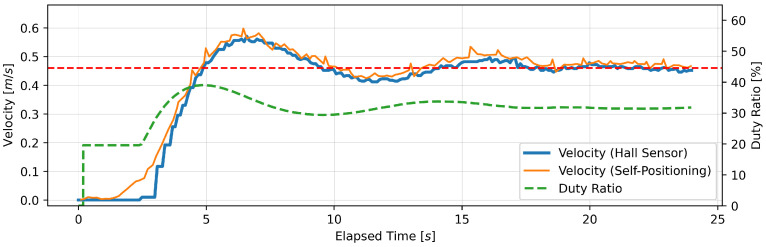
Output of speed and duty ratio at a load capacity of 14 kg and a target speed of 0.46 m/s.

**Figure 7 sensors-24-06456-f007:**
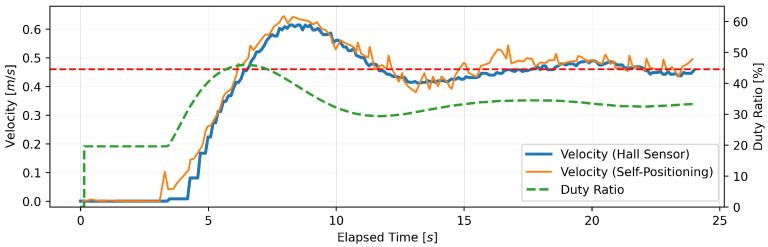
Output of speed and duty ratio at a load capacity of 20 kg and a target speed of 0.46 m/s.

**Figure 8 sensors-24-06456-f008:**
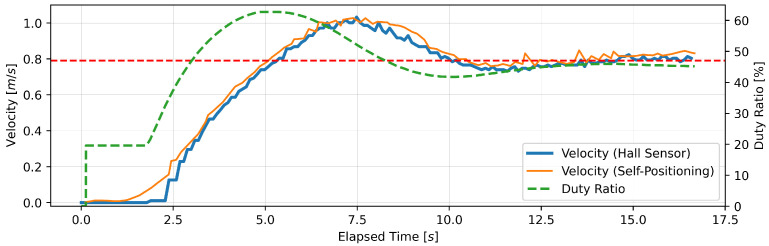
Output of speed and duty ratio at a load capacity of 8 kg and a target speed of 0.79 m/s.

**Figure 9 sensors-24-06456-f009:**
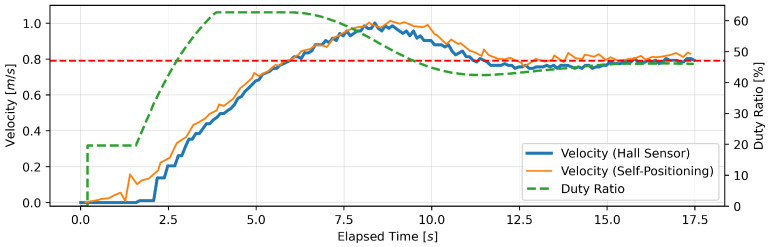
Output of speed and duty ratio at a load capacity of 14 kg and a target speed of 0.79 m/s.

**Figure 10 sensors-24-06456-f010:**
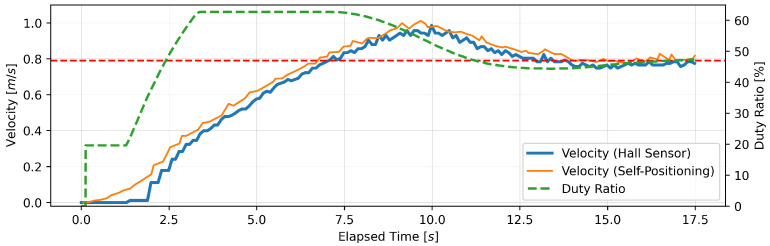
Output of speed and duty ratio at a load capacity of 20 kg and a target speed of 0.79 m/s.

**Figure 11 sensors-24-06456-f011:**
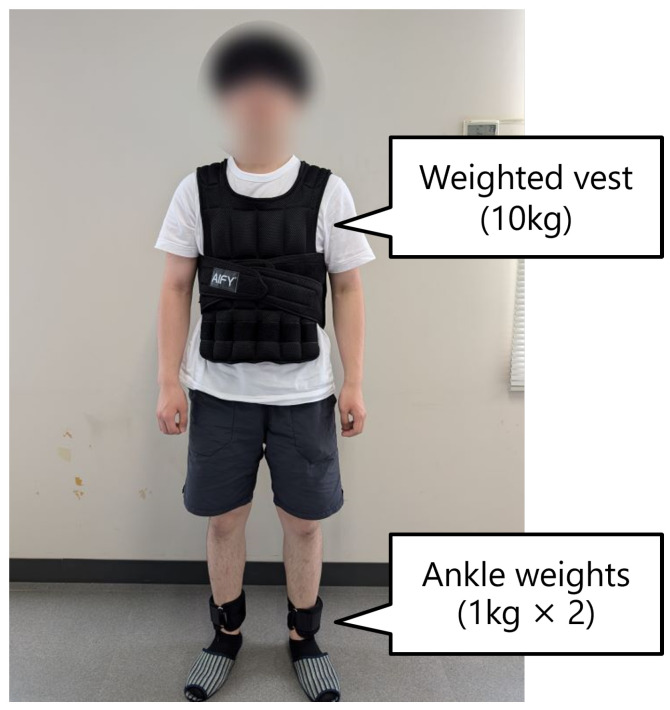
Subject with weights attached.

**Figure 12 sensors-24-06456-f012:**
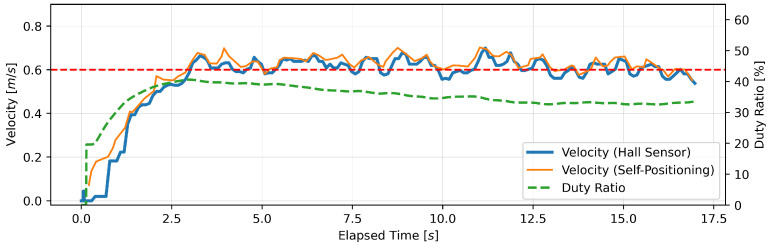
A single trial where walking speed stabilized at the target walking speed. The red dotted line represents Subject A’s average normal walking speed (0.601 m/s).

**Figure 13 sensors-24-06456-f013:**
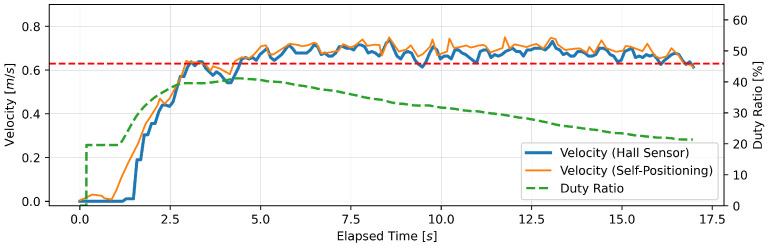
A trial where walking speed stabilized above the target speed. The red dotted line represents Subject C’s average normal walking speed (0.629 m/s).

**Figure 14 sensors-24-06456-f014:**
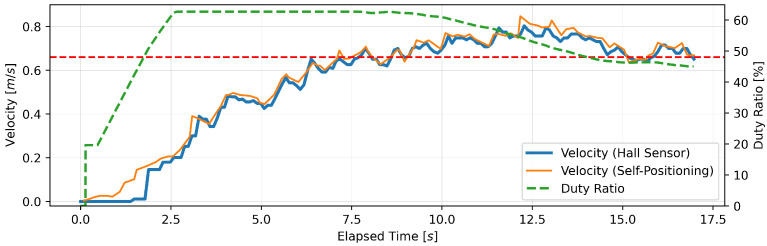
A trial with a slow initial speed. The red dotted line represents Subject E’s average normal walking speed (0.656 m/s).

**Figure 15 sensors-24-06456-f015:**
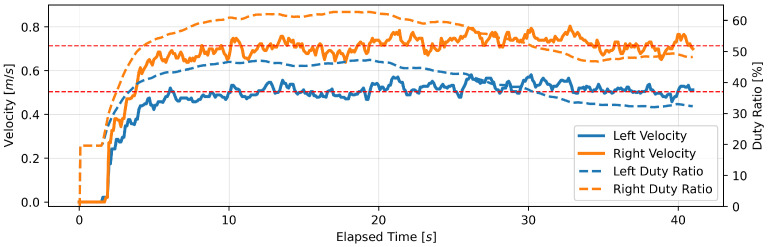
A trial of the circular route. Based on the pre-measurement results of Subject C, the ideal velocity for the left wheel was calculated to be 0.504 m/s, and for the right wheel 0.714 m/s.

**Table 1 sensors-24-06456-t001:** Devices mounted on a robotic walker.

Device	Number of Units
ZED 2 (Stereo camera)	1
NVIDIA Jetson Nano Developer Kit	1
Arduino Uno Rev3	1
Brushless DC Motor	2
Motor Driver (TB6605FTG)	2
Mobile Battery	2

**Table 2 sensors-24-06456-t002:** Questions to subjects after all trials.

No.	Question
Q1	During straight forward walking, I felt assistance in the same direction as my intended movement
Q2	During straight forward walking, variations in the amount of assistance did not interfere with my normal walking
Q3	During circular course walking, I felt assistance in the same direction as my intended movement
Q4	During circular course walking, variations in the amount of assistance did not interfere with my normal turning motion

**Table 3 sensors-24-06456-t003:** Statistics on walking speed when using a walker in the absence of support.

Subject Labels	Mean	SD	Min	Max
Subject A	0.601	0.034	0.538	0.657
Subject B	0.741	0.025	0.692	0.780
Subject C	0.629	0.015	0.603	0.668
Subject D	0.754	0.023	0.711	0.809
Subject E	0.656	0.028	0.597	0.704

**Table 4 sensors-24-06456-t004:** Subjects’ answers to the questions in Table 2.

Subject Labels	Q1	Q2	Q3	Q4
Subject A	Totally Agree	Totally Agree	Agree	Totally Agree
Subject B	Agree	Totally Agree	Neutral	Agree
Subject C	Totally Agree	Agree	Totally Agree	Totally Agree
Subject D	Totally Agree	Totally Agree	Totally Agree	Totally Agree
Subject E	Totally Agree	Agree	Totally Agree	Agree

## Data Availability

The original contributions presented in the study are included in the article, further inquiries can be directed to the corresponding authors.

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
