# Peer review of "Development of Assistance Level Adjustment Function for Variable Load on a Forearm-Supported Robotic Walker"

_sensors, 2024, doi:10.3390/s24196456_

Round 1

Reviewer 1 Report

Comments and Suggestions for Authors

This article focuses on the development of a support level adjustment function for variable loading on a forearm-supported robotic walker. Experiments were conducted with subjects to test the ability to maintain target speeds, and questionnaires confirmed that assistance did not interfere with actual walking. The proposed algorithm for determining levels of assistance demonstrated the ability to maintain target speeds and allowed the necessary level of assistance to be adjusted.

However, there are comments on the work:

1. The Abstract section should be rewritten to reflect the relevance of the problem solved and the scientific novelty of the solution obtained.

2. Keywords should be corrected by adding special terms characterizing the research.

3. How was determined and what value for the parameter number of degrees of freedom was obtained in the work?

4. What numerical characteristics of the reliability and safety of the walking robot for the patient were investigated in the work?

5. The list of cited sources should cite more current publications on strength and reliability research of systems and materials

6. Is it possible to set a specific temporal cycling of the patient's movement using a robot with motion tacting as in Google Fit?

7. The current limitations of the presented robot-assist control algorithm should be highlighted.

8. The conclusions should be structured by highlighting the main scientific and especially practical results obtained, as well as recommendations for designers of medical rehabilitation robots.

Reviewer 2 Report

Comments and Suggestions for Authors

This article clearly introduced its research objectives, which focus on the development of  a forearm-supported robotic walker with adjustable load-bearing capabilities. A walker support system based on a physical model and a PID control algorithm was proposed to provide consistent walking assistance to users under varying load conditions. This approach effectively addresses the limitations of traditional walkers, which do not adapt their assistance to dynamic changes in user load. The experimental procedure is presented concisely, emphasizing the rigor of hardware data acquisition. Importantly, the practicality of the employed algorithm is validated through real-world user trials, establishing a foundation for the clinical application of the walking aid.

Major comment:

Although the concept of incremental PID is not explicitly mentioned in the text, it is discernible from the equations presented that the chosen algorithm is indeed an incremental PID algorithm. However, the incremental PID algorithm, due to its focus solely on the most recent three sampling results, can lead to an integral truncation effect. The text fails to address whether this phenomenon may manifest differently under varying experimental conditions.

This article was submitted to a special issue called “Advances in IoT, AI and Sensor-Based Technologies for Disease Treatment, Health Promotion and Ageing Well”. After reviewing the article, I could hardly catch the correlation between the research and “Advances in IoT, AI and Sensor-Based Technologies”. Why did you choose to submit it to this special issue? I recommend that you describe the relevance of your research and this section in the begin of the article, so that readers can better understand the positioning of this article.

Minor comments:

1.      The literature review section briefly introduced studies related to intelligent walkers, yet it could benefit from a more extensive exploration, particularly focusing on the limitations of current walkers and the specific objectives of this study to address those limitations. Incorporating a comprehensive overview of the latest technological advancements in intelligent walkers, particularly those relevant to this research, would effectively underscore the novelty and distinctiveness of the present study.

2.      Although the article validated the effectiveness of the algorithm under various load conditions through experiments, the in-depth analysis of the experimental data is somewhat lacking. For instance, while the overshoot phenomena and response delays observed in certain scenarios are mentioned in the discussion, detailed cause analysis and improvement suggestions are not provided. Further analysis could potentially help explain the underlying reasons behind these phenomena and suggest possibilities for improving the algorithm.

3.      The current study, while employed only five young and healthy volunteers, inherently possesses limitations, because the target audience for this device is elderly individuals or those with mobility impairments. To improve the generalizability and applicability of the experimental findings, future research should endeavor to incorporate a more diverse and larger sample of target users, particularly the elderly, and conduct longer-term usage evaluations.

4.      The conclusion section can delve more specifically into the practical potential for improvement of the algorithm, as well as the challenges in future applications, such as integration with other technologies and further optimization of the algorithm.

Comments on the Quality of English Language

NULL

Round 2

Reviewer 1 Report

Comments and Suggestions for Authors

Overall, the article has been revised. The authors gave a detailed and comprehensive response to my comments. The article can be published in my opinion.

Reviewer 2 Report

Comments and Suggestions for Authors

NULL